# Personal relevance affects the perceived immorality of politically-charged threats

**Rebecca L. Dyer⬮\*, Nicklaus R. Herbst, Whitney A. Hintz, Keelah E. G. Williams**

Department of Psychology, Hamilton College, Clinton, New York, United States of America

\* rdyer@hamilton.edu

**Data Availability Statement:** All data files are available from the Open Science Framework (https://osf.io/62quz/).

**Funding:** The authors received funding for this research from the Hamilton College Psychology

## Abstract

Personal similarities to a transgressor makes one view the transgression as less immoral. We investigated whether personal relevance might also affect the perceived immorality of politically-charged threats. We hypothesized that increasing the personal relevance of a threat would lead participants to report the threat as more immoral, even for threats the participant might otherwise view indifferently. U.S. participants recruited online ($N = 488$) were randomly assigned to write about the personal relevance of either a liberal threat (pollution), conservative threat (disrespecting an elder), neutral threat (romantic infidelity), or given a control filler task. Participants then rated how immoral and personally relevant each political threat was, as well as reported their political ideology. Partial support for our hypothesis emerged: when primed with conservative writing prompts, liberal-leaning participants rated the conservative threat as more immoral, compared with the same threat after a liberal writing prompt. We did not find these results for conservative-leaning participants, perhaps because all participants cared relatively equally about the liberal threat.

## Introduction

In a polarized political climate, people tend to limit their exposure to opposing views. But what if people were made to consider the personal relevance of alternative perspectives? For example, Broockman and Kalla (2022) found that encouraging Fox News viewers to watch CNN for a month led to less polarized political attitudes [1]. Perhaps making alternative viewpoints seem more personally relevant affects the extent to which people interpret opposing viewpoints as threatening. Identifying how personal relevance affects moral threat perception can potentially inform attempts to gradually depolarize the political climate in the U.S.

The degree to which one views stimuli as threatening depends on features of the perceiver. For example, wearing a heavy backpack can make a slope seem steeper [2], and being alone can make a strange man seem more physically formidable than when one is in a group [3]. The extent to which one finds certain actions morally threatening may also depend on how psychologically close the action feels: the closer the threat, the more immoral the threat might seem. This effect may occur regardless of one's ideology. If moral threat perception is influenced by psychological closeness across political ideations, then manipulating psychological closeness might reduce political polarization. In the current paper, we explored how

Department (funding budgeted for faculty research, no specific grant/award). The funders had no role in study design, data collection and analysis, decision to publish, or preparation of the manuscript.

**Competing interests:** The authors have declared that no competing interests exist.

manipulating the perceived closeness (i.e., relevance) of politically-charged threats would affect the perceived immorality of those threats.

## Psychological distance and abstraction

People's perceptions of objects or events can be close or far away from the self, here, and now. This is the concept of *psychological distance*, which can be measured temporally, hypothetically, spatially, and socially [4]. Construal Level Theory considers how psychological distance influences cognition, postulating that the farther away a feature or event is the more abstract it will seem, whereas features or events that are closer seem more concrete [4, 5]. Abstract conceptualizations tend to be more general and less detail-oriented than concrete conceptualizations [6]. People are also more likely to successfully classify and recall concrete concepts as compared to abstract concepts [7].

Individuals are capable of thinking both abstractly and concretely, and abstraction can be manipulated via psychological distance [8]. For example, Liberman and Trope (1998) [5] manipulated psychological distance temporally by asking people to imagine themselves engaging in seven activities either "next year" (psychologically distant) or "tomorrow" (psychologically close) and found that high-level (i.e., abstract) responses were more common for participants in the psychologically distant condition, as compared to the psychologically close condition (e.g., participants perceived moving into a new apartment in terms of "starting a new life" rather than "packing and moving boxes"). Alternatively, Fujita et al. (2006) used an imagined scenario to manipulate spatial distance, finding that physically distant events were associated with more abstract thinking [9]. Researchers have also used visual tasks to manipulate abstraction, such as asking participants to focus on either the shape (abstract) or details (concrete) of a map [6]. Meta-analyses indicate that these different measures of psychological distance (temporal, spatial, hypothetical, etc.) have generally similar (though not necessarily equal) effects on abstraction [8, 10, 11, but see 12]. Because the level of abstraction changes the way people think about and approach phenomena, manipulating psychological distance can have downstream effects on varied behaviors, such as voting intentions, stereotyping, prosocial behavior, self-control, and creativity [4, 9].

## Psychological distance, morality, and politics

The way that people construe objects and events affects their moral judgment. In general, viewing moral violations as more temporally distant leads to harsher judgments [13], and future-oriented people show greater moral concern than present-oriented people [14]. This effect depends in part on the actor, however, as greater temporal distance actually leads to decreased moral condemnation of one's own behavior [15].

Thinking abstractly can also strengthen one's adherence to preexisting beliefs. For example, McCrea et al. (2012) found that participants were more likely to make stereotypic judgments about others when thinking abstractly (versus concretely) [16]. Similarly, people experience greater political polarization of particular issues when primed with an abstract mindset [17–19]. In one study, participants who were primed with abstract thinking were less influenced by situational factors and more likely to report attitudes in line with their reported political ideology [17]. This is likely because abstract thinking leads people to default to their core values. Priming participants to think concretely, on the other hand, can make them less polarized in their beliefs [19].

Moral threats that are relevant to the self are perceived as psychologically closer and concrete, whereas non-important threats are perceived as distant and abstract [20]. In this way, political partisans can mentally construe the same collective threat in an opposite manner. For

example, liberals view individualizing moral threats (such as threats to harm and fairness) as more concrete, and binding threats (such as threats to authority, ingroup, and purity) as more abstract; conservatives are the opposite [20, 21].

## Personal relevance and threat perception

Increased personal awareness can also increase our perception of threats. For example, Wormwood et al. (2016) tested the effects of general negative stimuli—images of the Boston Marathon bombings—on a shooting task [22]. This shooting task involved participants "shooting" armed targets and avoiding unarmed targets in a virtual setting. The researchers found that participants exposed to negative stimuli were more likely than the control group to shoot unarmed targets, showing a rise in threat sensitivity. For those in the Boston Marathon bombing condition, this effect was related to personal relevance: participants who said the Boston Marathon bombing was personally relevant to them were more likely to shoot unarmed people when primed with this negative imagery. This research indicates that personal relevance leads to a broader wariness of threats. Additionally, increased personal relevance can instill greater fear in people based on this enhanced threat awareness [23, 24]. When Weston (1996) gave participants messages about AIDS specific to their identity (an increase in concreteness and personal relevance) as opposed to messages unrelated to them, the participants were more likely to be fearful of AIDS and perceived their risk of getting AIDS as higher [24]. Messages that posed a personal threat to individuals increased their perception of personal risk and their fear of such threats.

Personal relevance can also influence one's judgments of others. When making decisions for oneself, as opposed to someone else, we tend to focus on concrete processes, rather than the abstract outcome itself [25]. A similar pattern occurs when making decisions for those who resemble us, as we are more likely to focus on these specific processes than we would for someone we do not know. Someone similar to us has ideas and interests personally relevant to us, so we think more concretely about them when making decisions. As a result, we tend to be more forgiving of transgressors who resemble ourselves [13] and we are harsher in judgment when we perceive personal similarities to a victim [26].

In the current study, we adopted personal relevance as our method of manipulating psychological distance. Personal relevance has been found to correlate with (and produce similar results to) many other measures of psychological distance, such as temporal, reality, spatial, and personal distance [27]. In the current context, we believed personal relevance would be an easily relatable concept for participants to grasp when considering different manners of political threats.

## Overview of the current study

Decreased psychological distance makes threats seem more relevant to the self. Transgressions against the self are judged more harshly than transgressions against others. This suggests that the more personally relevant a threat is, the more immoral that threat will seem. More concrete thinking is also related to less political polarization, suggesting that partisans may shift their perceived immorality of a political threat as a function of the perceived relevance of that threat. We therefore hypothesized that increasing the personal relevance of a politically-charged threat would also increase people's perceptions of the immorality of that threat, regardless of the participants' own political ideology. In other words, we hypothesized that increasing the personal relevance of a politically-charged threat would lead both conservative and liberal participants to report the threat as more immoral. This would occur even for threats the participant might otherwise view indifferently.

## Method

### Ethics statement

This research was approved by the Hamilton College Institutional Review Board under approval ID# F21-015. Written informed consent was obtained from all participants.

### Participants

A total of 455 participants completed an online survey during the spring of 2022. In our preliminary analysis we excluded two participants for failing an attention check, leaving us with 453 participants (219 men, 228 women, 5 non-binary, 1 preferred not to respond). Participants' ages ranged from 20–78 years old ($M$ = 42.91, $SD$ = 13.07) and largely identified as White (White: 75.3% (341/453), Black or African American: 7.1% (32/453), American Indian or Alaskan Native: 0.4% (2/453), Asian or Pacific Islander: 7.5% (34/453), Hispanic/Latino: 3.8% (17/453), Middle Eastern: 0.2% (1/453), Other: 0.4% (2/453), Multiple Ethnicities: 5.3% (24/453)). Participants in the United States were recruited through Amazon Mechanical Turk using CloudResearch (formerly TurkPrime [28, 29]). We used the CloudResearch Approved List to ensure high data quality, targeted a relatively equal number of participants across the political spectrum with CloudResearch's demographic options, and included three MTurk worker qualifications (HITs approved, Approval Rating, and location). The only inclusion criteria were that participants be at least 18 years of age and residing within the United States. Participants took a survey labeled "Personal Attitudes Study" and were compensated $0.50.

### Materials

**Personal relevance manipulation.** To manipulate the personal relevance of politically-charged threats, we gave participants one of four possible tasks. Prompts asked participants to "Please consider and write about how the following statement could be personally relevant to you"; participants were then given one of three statements: a liberal, conservative, or neutral threat. (Note that these labels were not presented to the participants; they were simply presented with the text of the threat.) The liberal write-up asked participants to "Imagine that a company lies about how much pollution they are causing." For the conservative write-up, we asked participants to "Imagine that someone disrespects an elder." For the neutral write-up, we asked participants to "Imagine that someone cheats on their romantic partner." The other priming task was a control condition in which participants were asked to list as many U.S. states and state capitals as possible. The liberal and conservative threats were developed for the current study by the experimenters through pilot testing to map onto issues generally associated with each political party (harm to the environment, disrespect towards authority [9]). Analyses of responses in the control condition confirmed that these two threats were viewed as more immoral by liberals and conservatives, respectively.

**Personal relevance and immorality measures.** Regardless of personal relevance condition, participants were asked to rate how personally relevant and immoral each threat (liberal, conservative, neutral) was using two Likert-type scales ranging from 1 (*Not at all*) to 7 (*Extremely*).

**Political ideology measure.** Participants self-reported their overall political ideology on a -50 (*very liberal*) to 50 (*very conservative*) sliding scale.

### Design and procedure

Participants were given a written consent form and told that this was a five-minute study about personal attitudes (participants indicated their consent by typing in their initials, but no

other identifying information was collected). Next, participants were randomly assigned to one of the four personal relevance write-up conditions and given two minutes to complete the task. Following the personal relevance manipulation, participants were asked to respond to a series of statements. We then presented participants with the liberal, conservative, and neutral threat and asked participants to rate each the personal relevance and immorality of each of these threats. The threats were presented in a random order. Participants then responded to demographic measures (age, sex, race/ethnicity) and were debriefed and thanked for their time.

## Statistical analysis

We used IBM SPSS Statistics 27 to conduct our statistical analysis. To conduct our analyses on political ideology and personal relevance we first created a political split variable and coded all responses to the political ideology slider scale that were less than zero as -1 (liberal) and all responses greater than zero as 1 (conservative). All data are available at: https://osf.io/62quz/.

## Results

### Preliminary analyses

We first explored the overall relationship between our focal variables, and found significant positive correlations between all Likert items. Most importantly, some of the strongest correlations were between each political threat's (i.e., liberal, conservative) personal relevance rating and the corresponding immorality rating (see Table 1). For example, the personal relevance of the liberal threat (a company lying about their pollution) was significantly positively correlated with the perceived immorality of that threat, $r = .43$, $p < .001$. The personal relevance of the neutral threat was also positively correlated with the perceived immorality of the neutral threat, although this effect was smaller, $r = .26$, $p < .001$. These findings indicate that across all four conditions there were significant positive relationships between how personally relevant a participant found a threat and how immoral they found that threat to be.

### Primary analyses

**Personal relevance.**  Next, we conducted a one-way analysis of variance (ANOVA) on the personal relevance rating as a function of personal relevance write-up condition. For the conservative personal relevance rating, this relationship was statistically significant, $F(3, 449) = 12.64$, $p < .001$. The strength of the relationship, as indexed by $eta^2$, was .078, indicating a medium effect. An independent samples $t$-test comparing the conservative condition to all

**Table 1. Correlations between personal relevance and immorality ratings.**

| Variables | CIPR | CII | LIPR | LII | NIPR | NII | *M* | *SD* |
|---|---|---|---|---|---|---|---|---|
| **CIPR** | 1 | | | | | | 4.47 | 1.95 |
| **CII** | .540[a] | 1 | | | | | 5.19 | 1.61 |
| **LIPR** | .360[a] | .154[a] | 1 | | | | 4.87 | 1.89 |
| **LII** | .136[a] | .336[a] | .426[a] | 1 | | | 5.97 | 1.35 |
| **NIPR** | .472[a] | .249[a] | .340[a] | .150[a] | 1 | | 4.54 | 2.18 |
| **NII** | .236[a] | .453[a] | .131[a] | .454[a] | .255[a] | 1 | 6.23 | 1.17 |

CIPR, Conservative Item Personal Relevance; CII, Conservative Item Immorality; LIPR, Liberal Item Personal Relevance; LII, Liberal Item Immorality; NIPR, Neutral Item Personal Relevance; NII, Neutral Item Immorality. *N* = 453.

[a] Correlation is significant at the $p < 0.01$ level (2-tailed)

other conditions indicated that mean conservative personal relevance ratings were significantly higher for participants in the conservative condition ($M = 5.40$, $SD = 1.56$) than those in all other conditions ($M = 4.16$, $SD = 1.97$), $t(451) = -6.83$, $p < .001$, 95% CI [-1.64, -.084]. This indicates that the conservative personal relevance manipulation made people view the conservative threat as more personally relevant.

For the liberal personal relevance rating, we found no main effect of condition, $F(3, 449) = 1.23$, $p = .298$. However, when completing an independent samples $t$-test of the liberal condition compared to all other conditions, there was marginal significance, $t(451) = -1.78$, $p = .077$. This indicates that participants who were in the liberal write-up condition viewed the liberal threat as somewhat more personally relevant ($M = 5.14$, $SD = 1.73$) compared to other groups ($M = 4.79$, $SD = 1.94$), 95% CI [-0.75,0.06].

For the neutral relevance rating, we found no effect of condition on perceived relevance, $F(3, 449) = 0.99$, $p = .397$. There was also no significant effect when comparing those in the neutral write-up to all other conditions, $t(451) = -1.44$, $p = .152$. Writing about the personal relevance of an infidelity threat did not lead participants to view that threat as more personally relevant.

**Immorality ratings.**   We conducted a one-way ANOVA on conservative threat immorality ratings as a function of personal relevance write-up condition. This relationship was marginally significant, $F(3, 449) = 2.61$, $p = .051$. The strength of the relationship, as indexed by eta$^2$, was .017, indicating a small effect. An independent samples $t$-test indicated that mean conservative threat immorality ratings were marginally higher for participants in the conservative condition ($M = 5.54$, $SD = 1.42$) than all other groups ($M = 5.08$, $SD = 1.65$), $t(451) = -2.68$, $p = .008$, 95% CIs [-0.81, -0.12]. Therefore, writing about the personal relevance of a conservative threat caused participants to rate the conservative threat as more immoral as compared to writing about any other threat. In contrast, there was no effect of personal relevance write-up condition on people's liberal immorality threat ratings, $F(3, 449) = 0.52$, $p = .670$. There was also no significant effect of condition when comparing participants in the liberal condition to all other conditions, $t(451) = 1.25$, $p = .214$. Writing about the personal relevance of a company lying about pollution did not lead participants to rate that threat as more immoral.

We then ran independent-samples $t$-tests comparing participants in the neutral condition to all other participants and found that participants in the neutral condition didn't see the neutral threat as any more immoral than participants in any other condition, $t(451) = 0.25$, $p = .805$. We suspect these findings may be due to a ceiling effect for the neutral threat, because everyone seemed to find romantic infidelity to be highly immoral (rating this threat in the 6 range on a scale of 1–7).

**Political ideology and the role of personal relevance.**   Our next hypothesis was that increasing the personal relevance of a politically-charged threat would lead both conservative and liberal participants to report the threat as more immoral. This would occur even for threats the participant might otherwise view indifferently. To test this hypothesis, we split the sample based on participant self-reported political ideology on a -50 (*very liberal*) to 50 (*very conservative*) sliding scale. Responses greater than zero were coded as conservative, while responses less than zero were coded as liberal. Participants who reported exactly zero were excluded from this portion of the analysis.

First, we looked solely at how liberals viewed the conservative threat after the conservative write-up condition, as compared to other conditions. An independent-samples $t$-test compared whether liberal participants who did and did not complete the conservative write-up differed in how immoral they rated the conservative Likert items. The test was statistically significant, $t(158) = -2.51$, $p = .013$, $d = -0.45$, indicating that liberals in the conservative write-

up condition saw the conservative threat as more immoral ($M$ = 5.37, $SD$ = 1.36) than liberals in any other condition ($M$ = 4.67, $SD$ = 1.65); 95% CI [-1.26, -0.15]. A second $t$-test analyzing personal relevance was also significant, $t(158)$ = -4.68, $p < .001$, $d$ = -0.79, and showed that liberals in the conservative write-up condition saw the conservative threat as more personally relevant ($M$ = 5.00, $SD$ = 1.60) than liberals in any other condition ($M$ = 3.62, $SD$ = 1.80); 95% CI [-1.97, -0.80].

We then explored whether conservatives viewed the liberal threat as more immoral and personally relevant after completing the liberal write-up condition, as compared to other conditions. An independent-samples $t$-test compared whether conservative participants who did and did not complete the liberal write-up differed in how immoral and personally relevant they reported feeling about the liberal Likert items. The test for immorality was not statistically significant, $t(248)$ = 1.53, $p$ = .133, indicating that conservatives in the liberal write-up condition did not see the liberal threat as significantly more immoral ($M$ = 5.94, $SD$ = 1.15) than conservatives in any other condition ($M$ = 5.85, $SD$ = 1.41); 95% CI [-0.47, 0.30]. However, conservatives in the liberal write-up condition did see the liberal threat as marginally significantly more personally relevant ($M$ = 5.08, $SD$ = 1.76) as compared to conservatives in other conditions ($M$ = 4.55, $SD$ = 2.07), $t(248)$ = -1.96, $p$ = .052, $d$ = -0.07; 95% CI [-1.05, 0.00].

**Exploratory analyses.** We did not have any *a priori* hypotheses regarding the role of participant demographic variables. However, we decided to conduct an exploratory analysis to investigate whether any of these variables might moderate our effects. None of the primary analyses reported above were affected by participant gender or age. We were unable to explore the effects of participant race/ethnicity due to insufficient variability in our sample.

## Discussion

Consistent with our primary hypothesis, we found that personal relevance was significantly positively associated with the perceived immorality of politically-charged threats. Through manipulating personal relevance, we found partial support for a causal relationship, such that increasing the personal relevance of a conservative threat marginally increased the perceived immorality of that threat. We did not see a similar effect of personal relevance on perceived immorality of the liberal and neutral threats.

When liberal participants wrote about the personal relevance of disrespecting an elder, they then increased their perceived immorality of that action. Thus, a novel contribution of this study was making liberal-identifying participants see an otherwise "conservative" threat as more immoral when asked to write about the personal relevance of the threat. By manipulating personal relevance, we find some evidence that the personal relevance of a threat guides our perceptions of the immorality of that threat.

Although liberals in the conservative write-up condition found the conservative threat more personally relevant and more immoral, conservatives in the liberal write-up condition saw the liberal threat as marginally more personally relevant, but not more immoral. Conservatives may not have changed their immorality ratings of the liberal threat after seeing it as more personally relevant for multiple reasons. First, the wording of the liberal threat (i.e., "Imagine that a company lies about how much pollution they are causing") may have allowed them to focus on the lying aspect of the threat rather than the pollution aspect. For example, one participant in the liberal write-up condition wrote, "This angers me and I think the company should be punished both financially and criminally. People who put their interests above others who have no say should be held accountable." Lying and cheating are not polarized political issues, and therefore all participants may have cared relatively equally about the alleged "liberal threat" to begin with. Indeed, participants in the control condition ($M$ = 5.88,

SD = 1.39) found the liberal threat to be no less immoral than participants in the liberal write-up condition (M = 5.94, SD = 1.95), t(131) = -.24, p = .810.

Previous research suggests that emotions might mediate the relationship between political threat perception and support for relevant policies. In a series of three experiments, Eadeh and Chang (2020) found that a liberal threat increased support for relevant liberal policies, and this effect was mediated by anger [30]. Thus, future research should consider the particular emotions elicited by politically-charged threats, and whether such emotions guide the perceived immorality of these threats.

Our findings cohere with existing literature on the effects of psychological distance on moral decision-making. In studies in America and the Netherlands, researchers have found that people judge violations more severely when they involve the self [31, 32]. As we find in the present study, the more concrete a violation is, the more immoral it becomes. Similar results are also seen in Eastern cultures. Zhang et al. (2022), for example, examined the effects of psychological distance on moral judgments in their population of undergraduates in China [33]. Manipulating spatial and temporal distances of morally-relevant acts across two experiments, the researchers found that closer psychological distance increases perceptions of immorality. Thus, by making the threat close—more concrete—people viewed the threat as more immoral regardless of their traditional core beliefs, making them more willing to accept other perspectives. One can see psychological distance's effect on immorality beyond the American population of our study.

Future researchers exploring similar questions might consider varying how they manipulate personal relevance. For instance, further research may want to explore more subliminal ways of inducing personal relevance such as using more second person language. Additionally, future studies can examine the long-term effects of a personal relevance manipulation and the impact of constant exposure [e.g., 1] as compared to a singular exposure (e.g., our personal write-up manipulation). Future research can also account for individual differences across threats by testing a variety of politically-charged threats rather than only one for each political orientation. Further, political parties are not a monolith, and other third variables such as geographic location (e.g., rural vs. urban) may explain the nature of someone's particular political opinions and ideology better than their political affiliation.

Overall, our results support a significant positive relationship between personal relevance and immorality. We found that psychological distance plays a role in the judgment of political threats. These findings also expand upon previous literature, showing that when liberals see conservative threats as more personally relevant, they are capable of seeing those threats as more immoral. This research is important to understanding the factors that contribute to forming political ideologies. Additionally, this study shows that ideologies are not stagnant: they have the ability to shift through various components of psychological distance, namely personal relevance.

## Author Contributions

**Conceptualization:** Rebecca L. Dyer, Nicklaus R. Herbst, Whitney A. Hintz, Keelah E. G. Williams.

**Formal analysis:** Rebecca L. Dyer, Nicklaus R. Herbst, Whitney A. Hintz, Keelah E. G. Williams.

**Funding acquisition:** Rebecca L. Dyer, Keelah E. G. Williams.

**Investigation:** Rebecca L. Dyer, Nicklaus R. Herbst, Whitney A. Hintz, Keelah E. G. Williams.

**Methodology:** Rebecca L. Dyer, Nicklaus R. Herbst, Whitney A. Hintz, Keelah E. G. Williams.

**Project administration:** Rebecca L. Dyer, Keelah E. G. Williams.

**Supervision:** Rebecca L. Dyer, Keelah E. G. Williams.

**Writing – original draft:** Rebecca L. Dyer, Nicklaus R. Herbst, Whitney A. Hintz, Keelah E. G. Williams.

**Writing – review & editing:** Rebecca L. Dyer, Nicklaus R. Herbst, Whitney A. Hintz, Keelah E. G. Williams.

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
