## [Decision Letter · Decision Letter 0]

20 Sep 2023

PONE-D-23-10190Personal relevance affects the perceived immorality of politically-charged threatsPLOS ONE

Dear Dr. Dyer,

Thank you for submitting your manuscript to PLOS ONE. After careful consideration, we feel that it has merit but does not fully meet PLOS ONE’s publication criteria as it currently stands. Therefore, we invite you to submit a revised version of the manuscript that addresses the points raised during the review process.

We look forward to receiving your revised manuscript.

Kind regards,

Larissa M. Batrancea

Academic Editor

PLOS ONE

Journal Requirements:

"We would like to thank the Hamilton College Psychology Department for their support and for funding this research."

"The authors received funding for this research from the Hamilton College Psychology Department (funding budgeted for faculty research, no specific grant/award). The funders had no role in study design, data collection and analysis, decision to publish, or preparation of the manuscript."

Additional Editor Comments:

Dear authors,

Please follow the reviewers' instructions in order to improve the content of the manuscript.

Reviewers' comments:

Reviewer's Responses to Questions

**Comments to the Author**

1. Is the manuscript technically sound, and do the data support the conclusions?

Reviewer #1: Partly

Reviewer #2: Partly

Reviewer #3: Yes

2. Has the statistical analysis been performed appropriately and rigorously? 

Reviewer #1: Yes

Reviewer #2: No

Reviewer #3: Yes

3. Have the authors made all data underlying the findings in their manuscript fully available?

Reviewer #1: Yes

Reviewer #2: No

Reviewer #3: Yes

4. Is the manuscript presented in an intelligible fashion and written in standard English?

Reviewer #1: Yes

Reviewer #2: Yes

Reviewer #3: Yes

5. Review Comments to the Author

Reviewer #1: This is a novel study which has found that personal relevance does influence how individuals perceive politically charged threats as moral or immoral. The study findings are quite relevant and insightful given the current political developments across the globe. The manuscript is well-written especially the introduction section can be understood even by readers from other disciplines.

To improve the manuscript further, I have following suggestions:

A.)In the sub section titled 'participant' under Methods section, I would suggest the following changes:

1. Please describe briefly the inclusion and exclusion criterias which were used to select the participants.

2.A table will be a better of representing the characteristics of the participants. Please add a Table there.

B.)What was the basis of selection of the particular write up question for each of the categories conservative,liberal and neutral.

C.)Table-1, the results section where preliminary analyses is discussed:

The correlation between NII and NIPR seems to be low as opposed to what is observed in case of liberal and conservative threats. Discuss a bit about it in the preliminary analyses section.

D.) The discussion section currently describes more of the study findings and doesn't compare and contrasts with the findings of similar studies conducted in U.S and elsewhere. Inclusion of recent literature would highlight the importance of your study. So, I would suggest the inclusion of some relevant and current literature in the discussion section.

E.) A query related to the study is that age, sex and ethnic origins might also be influencing the link that you are studying, how do you adjust for these factors in your study.

Reviewer #2: This paper studies an interesting topic about revealed political preferences, accounting for personal relevance to identify how immoral are the threats. The strategy is to identify how political ideology is linked to more immoral threats. The idea is not completely clear. The authors must estate the importance of identifying these effects. Is it to understand a polarized society? If then, why polarization can be viewed as important for society? Perhaps this effect seems obvious, but the authors can be more precise about the relevance of this effect.

There are major changes that are recommended to improve the paper. There is no data analysis about the participants other than their ethnicity differentiation. How young are the participants? Is it in a US state where the political preferences are identified? how is this related to the main question of the authors?

How the definitions of liberal, conservative, or neutral threat are defined for everyone? Or does it depend on how people understand these ideologies? It would be helpful to know the density of the answers before showing the variables' correlation. I do not think an ANOVA analysis is sufficiently robust to answer the questions. I would recommend that authors consider probabilistic models to estimate the conditional probability of a person declaring their political orientation and their evaluation of the threat of immortality. Perhaps, a multinomial model can help to account for the possibility that a person ranks differently from the alternatives.

Reviewer #3: This manuscript has a very current, interesting theme with global sociopolitical impact. In the face of a polarized world, presenting a scientific perspective for the conservative and liberal perception regarding moral assessments sheds light on a very relevant topic.

It provides an academic tool to bring together different views in the liberal versus conservative context.

The study was well designed methodologically, the statistical analysis used was appropriate.

The basis of the literature is correct. The conclusions reflect the results presented.

I agree with the publication of the manuscript.

6. PLOS authors have the option to publish the peer review history of their article (what does this mean?). If published, this will include your full peer review and any attached files.

Reviewer #1: No

Reviewer #2: No

Reviewer #3: **Yes: **Priscila C Andrade

---

## [Author Response · Author response to Decision Letter 0]

3 Nov 2023

PLOS ONE Requirements:

1. We have verified that the manuscript follows all of PLOS ONE’s style requirements.

2. We have removed the reference to funding information from the Acknowledgements in the manuscript. The funding statement (“The authors received funding for this research from the Hamilton College Psychology Department (funding budgeted for faculty research, no specific grant/award). The funders had no role in study design, data collection and analysis, decision to publish, or preparation of the manuscript.”) can remain as is.

3. We added a new “Ethics Statement” section to the manuscript, including the full ethics statement. This section can be found at the very beginning of the Method section (lines 139-141 in the clean version).

4. The data are currently available through the Open Science Framework (https://osf.io/62quz/). We have added a sentence to the manuscript to make this clearer (lines 195-196).

5. We have verified that the reference list is complete and correct. None of the cited papers have been retracted.

Reviewer 1’s Comments:

● This is a novel study which has found that personal relevance does influence how individuals perceive politically charged threats as moral or immoral. The study findings are quite relevant and insightful given the current political developments across the globe. The manuscript is well-written especially the introduction section can be understood even by readers from other disciplines.

We thank Reviewer 1 for their positive assessment of our research and this manuscript. We have considered each of R1’s suggestions in turn:

● A.) In the sub section titled 'participant' under Methods section, I would suggest the following changes: 1. Please describe briefly the inclusion and exclusion criterias which were used to select the participants. 2.A table will be a better of representing the characteristics of the participants. Please add a Table there.

We added some clarifying information in the Participants section, specifying that all participants were required to be at least 18 years old and be living in the United States (lines 154-155). In terms of adding a table to represent the participant demographics, we respectfully disagree with the suggestion. We have followed APA convention for reporting demographics and do not believe that a table would be an appropriate way to present this information.

● B.) What was the basis of selection of the particular write up question for each of the categories conservative, liberal and neutral.

We have modified language within the manuscript to specify that the experimenters in the current study devised these materials based on pilot testing (lines 168-171).

● C.) Table-1, the results section where preliminary analyses is discussed:

The correlation between NII and NIPR seems to be low as opposed to what is observed in case of liberal and conservative threats. Discuss a bit about it in the preliminary analyses section.

Thank you for pointing out this feature of our results. We have added language that clarifies that the strongest associations are between the political threats’ personal relevance rating and immorality rating, and we have highlighted the weaker nature of the neutral threat’s personal relevance and immorality rating correlation (lines 201, 204-206). However, although the correlation is weaker, it is still statistically significant at p < .001 and in the same direction as that found with the political threats. 

● D.) The discussion section currently describes more of the study findings and doesn't compare and contrasts with the findings of similar studies conducted in U.S and elsewhere. Inclusion of recent literature would highlight the importance of your study. So, I would suggest the inclusion of some relevant and current literature in the discussion section.

We are grateful for this helpful feedback and have added significantly more interpretation in light of existing research to our Discussion section. See lines 327-338.

● E.) A query related to the study is that age, sex and ethnic origins might also be influencing the link that you are studying, how do you adjust for these factors in your study.

We appreciate the suggestion to consider individual difference variables as moderators for our effects. We did not address this in the original paper as we do not have a priori hypotheses about how age or sex might influence our results, and, unfortunately, we do not have sufficient variability in race/ethnicity to be able to explore its potential effects. In response to this comment, we revisited our data and ran exploratory analyses to determine whether age and gender moderate our primary results. They do not. We have added an “Exploratory analyses” section to the paper to note these findings. See lines 288-293.

We thank Reviewer 1 again for their helpful comments.

Reviewer 2’s Comments:

● A) This paper studies an interesting topic about revealed political preferences, accounting for personal relevance to identify how immoral are the threats. The strategy is to identify how political ideology is linked to more immoral threats. The idea is not completely clear. The authors must estate the importance of identifying these effects. Is it to understand a polarized society? If then, why polarization can be viewed as important for society?

We appreciate the suggestion to emphasize for the reader why our research is important. To do so, we have added several lines of text to the Introduction (see lines 36-37) to directly connect manipulating psychological closeness to potential depolarization efforts. We also added lines 43-45 where we emphasize that we are exploring how manipulating personal relevance can be more intentionally utilized to depolarize our political climate if it significantly impacts perceptions of moral threats. We also note that our discussion section ends with a concluding statement about the importance of our findings (lines 353-356).

● B) There is no data analysis about the participants other than their ethnicity differentiation. How young are the participants? Is it in a US state where the political preferences are identified? how is this related to the main question of the authors?

We provide information about participant demographics on lines 145-150. This includes information about participants’ gender, age, and race/ethnicity. We now specify that participants were living in the United States at the time of data collection. However, we did not collect information about participants’ specific location either in terms of U.S. state of residence or zip code. Information about participants’ ideologies was assessed at an individual level by asking participants to report their overall political ideology on a -50 (very liberal) to 50 (very conservative) sliding scale. Participants’ political ideologies are related to the main research question because we wanted to demonstrate that we could shift both conservative and liberal participants’ perception of politically-charged threats. We state this goal explicitly in lines 134-137.

● C) How the definitions of liberal, conservative, or neutral threat are defined for everyone? Or does it depend on how people understand these ideologies? It would be helpful to know the density of the answers before showing the variables' correlation.

We appreciate the opportunity to clarify this point. We did not define our threats as “liberal”, “conservative”, or “neutral” to the participant. Instead, participants were simply presented with one of the threats themselves (a company lying about pollution, a person disrespecting an elder, or a person committing infidelity). We, the researchers, had categorized these threats as liberal, conservative, or neutral based on pilot testing and previous research. To clarify this point, we have added a parenthetical on lines 162-163 noting that the ideological labels were never seen by the participants. We believe that this misunderstanding led to the reviewer’s comment about the density of the answers, which is now moot. 

● D) I do not think an ANOVA analysis is sufficiently robust to answer the questions. I would recommend that authors consider probabilistic models to estimate the conditional probability of a person declaring their political orientation and their evaluation of the threat of immortality. Perhaps, a multinomial model can help to account for the possibility that a person ranks differently from the alternatives.

While we appreciate the suggestion, we respectfully disagree. The ANOVA we conducted is the appropriate statistical test under the circumstances, and we note that both Reviewer 1 and Reviewer 3 agree that the analyses were “performed correctly and rigorously.” Indeed, Reviewer 3 specifically states, “the statistical analysis used was appropriate” in their comments. 

We appreciate the thorough and thoughtful review that Reviewer 2 has conducted of our paper, and believe that we have fully addressed their concerns. 

Reviewer 3’s Comments:

● A) This manuscript has a very current, interesting theme with global sociopolitical impact. In the face of a polarized world, presenting a scientific perspective for the conservative and liberal perception regarding moral assessments sheds light on a very relevant topic. It provides an academic tool to bring together different views in the liberal versus conservative context.

We thank Reviewer 3 for their kind and positive assessment of our work.

● B) The study was well designed methodologically, the statistical analysis used was appropriate. The basis of the literature is correct. The conclusions reflect the results presented. I agree with the publication of the manuscript.

We again thank Reviewer 3 for their thoughtfulness and time.

---

## [Editor Report · Decision Letter 1]

8 Dec 2023

Personal relevance affects the perceived immorality of politically-charged threats

PONE-D-23-10190R1

Dear Dr. Dyer,

We’re pleased to inform you that your manuscript has been judged scientifically suitable for publication and will be formally accepted for publication once it meets all outstanding technical requirements.

Kind regards,

Larissa M. Batrancea

Academic Editor

PLOS ONE

---

## [Editor Report · Acceptance letter]

18 Dec 2023

PONE-D-23-10190R1 

PLOS ONE

Dear Dr. Dyer, 

I'm pleased to inform you that your manuscript has been deemed suitable for publication in PLOS ONE. Congratulations! Your manuscript is now being handed over to our production team.

Kind regards, 

on behalf of

Dr. Larissa M. Batrancea 

Academic Editor

PLOS ONE